# Profile Likelihood for Hierarchical Models Using Data Doubling

**DOI:** 10.3390/e25091262

**Published:** 2023-08-25

**Authors:** Subhash R. Lele

**Affiliations:** Department of Mathematical and Statistical Sciences, University of Alberta, Edmonton, AB T6G 2R3, Canada; slele@ualberta.ca

**Keywords:** data cloning, functions of parameters, Laplace approximation, nuisance parameters, parameterization invariance

## Abstract

In scientific problems, an appropriate statistical model often involves a large number of canonical parameters. Often times, the quantities of scientific interest are real-valued functions of these canonical parameters. Statistical inference for a specified function of the canonical parameters can be carried out via the Bayesian approach by simply using the posterior distribution of the specified function of the parameter of interest. Frequentist inference is usually based on the profile likelihood for the parameter of interest. When the likelihood function is analytical, computing the profile likelihood is simply a constrained optimization problem with many numerical algorithms available. However, for hierarchical models, computing the likelihood function and hence the profile likelihood function is difficult because of the high-dimensional integration involved. We describe a simple computational method to compute profile likelihood for any specified function of the parameters of a general hierarchical model using data doubling. We provide a mathematical proof for the validity of the method under regularity conditions that assure that the distribution of the maximum likelihood estimator of the canonical parameters is non-singular, multivariate, and Gaussian.

## 1. Introduction

Hierarchical models, also known variously as latent structure models, mixture models, multilevel models, mixed effects models, etc., are an important class of models with substantial scientific applications [1]. The problems of missing data and measurement error, either in covariates or in observations, are also common in practice. This can be handled appropriately through hierarchical models. Statistical inference for these models is difficult, because the likelihood function involves high-dimensional integration over the latent, unobserved variables. The development of the Markov chain Monte Carlo (MCMC) algorithms and related methods have facilitated statistical inference for these models using the fully Bayesian approach [1]. The Bayesian approach with non-informative, flat priors is commonly used to conduct statistical inference for hierarchical models. However, frequentists argue against the use of flat priors because they do not always lead to proper posterior distributions, hence throwing doubt on their utility in scientific inference. Flat priors are also problematic because a prior that is flat on one scale is not necessarily flat on any other scale [2,3], thus calling into question their ’non-informativeness’. A natural consequence is that flat priors on the canonical parameters imply informative prior distribution on derived parameters [4]. The practical consequences of the non-invariance to parameterization are discussed in [5,6]. The non-invariance of the Bayesian inference to the parameterization of the model is considered scientifically problematic. In contrast to this, likelihood-based frequentist inference is invariant to the parameterization of the model. Likelihood-based inference for hierarchical models can be conducted using various methods, such as expectation-maximization (EM), Monte Carlo EM, and the Monte Carlo Newton Raphson method [7]. An alternate method of data cloning (DC) [8] can be applied by the minor modification of the standard Bayesian models using commonly available Bayesian computational tools such as JAGS [9] or Stan [10]. Instead of the MCMC algorithm, integrated Laplace approximation (INLA) [11] uses an analytical approach to obtain approximate posterior distributions but only for a restricted set of models. It appears that the estimation of the canonical parameters of a general hierarchical model can be handled in many different ways, each with its own pros and cons.

In many scientific problems, quantities of interest are often specified functions of the canonical parameters of the model. For example, in population dynamics models, one may be interested in the probability of extinction, which is a real-valued function of the parameters of the model [12]. In epidemiologic studies, the basic reproduction number $R_{0}$ is a real-valued function of the parameters of a system of differential equations [13]. Frequentist inference for a specified function of parameters is usually conducted using the profile likelihood [14,15,16]. Similarly to the likelihood function, the profile likelihood function is invariant to the parameterization of the model. Unfortunately, similarly to computing the likelihood function, computing the profile likelihood for a general hierarchical model is difficult. In this paper, we extend the method of data doubling (DD) suggested by [17] to compute the profile likelihood function for any specified function of the canonical parameters of a general hierarchical model.

Past attempts to compute the profile likelihood for a general hierarchical model have used the well-known Geyer–Thompson algorithm [18] to obtain the MCMC estimate of the likelihood ratio. The algorithm in [19] is useful for obtaining the profile likelihood for a *component* of the canonical parameter vector and not for a general function of the canonical parameters. This method also requires the repeated use of the DC algorithm, making it computationally expensive. Alternatively, [20] estimates the entire likelihood surface and then uses constrained optimization to obtain the profile likelihood for a specified function of the parameters. However, this procedure is difficult to implement if the number of parameters in the model is large. Our method, on the other hand, is closely related to [17], where it is shown that one can obtain the profile likelihood and adjusted profile likelihood for a general function of a parameter using data doubling and non-parametric bootstrapping. Unfortunately, it is difficult to bootstrap general hierarchical models; the computational cost can be prohibitive, and there is generally no unique way to resample the data. In the present paper, we show that one can obtain the profile likelihood using data doubling and *posterior* distributions. Estimating the posterior distribution of any specified function of the canonical parameters is computationally easy. Our method thus increases the application scope of profile likelihood inference to general hierarchical models.

## 2. Bayesian Posterior, Profile Likelihood, and Data Doubling

Below, we introduce some mathematical notation:Let Y∼f(y;θ) be the probability density function of *Y* (real- or vector-valued random variable) with canonical parameters θ. We assume that the dimension of θ is d>1.Let ψ=f(θ) be a real-valued function of the canonical parameters that is of scientific interest.Let θ↔(ψ,λ) be a one-to-one transformation of the canonical parameters. Parameter ψ is of scientific interest, and the incidental parameters are λ. The incidental parameters λ may be vector-valued.Let π(θ) denote the prior distribution on the canonical parameters. With a slight misuse of notation, let us denote the induced prior distribution on (ψ,λ) as π(ψ,λ)=π(ψ)π(λ|ψ).The log-likelihood function is l(ψ,λ;y)=logf(y;ψ,λ), where y is the observed (hence, fixed) data. This is a function of the parameters (ψ,λ).The negative Hessian matrix is denoted by $i\left(\psi ,\lambda \right)=-\left(\begin{array}{cc} \frac{\partial^{2}l}{\partial \psi^{2}} & \frac{\partial^{2}l}{\partial \psi \partial \lambda }\\ \frac{\partial^{2}l}{\partial \lambda \partial \psi } & \frac{\partial^{2}l}{\partial \lambda^{2}} \end{array}\right)$. We write the component of this matrix that corresponds to the nuisance parameters λ as $i_{\lambda \lambda }\left(\psi ,\lambda \right)=\left(\frac{\partial^{2}l}{\partial \lambda^{2}}\right)$. This component is a square (sub)matrix of dimension (d−1).Let us write the inverse of the Hessian matrix as $i^{-1}=\left(\begin{array}{cc} i^{\psi \psi } & i^{\psi \lambda }\\ i^{\lambda \psi } & i^{\lambda \lambda } \end{array}\right)$.The log-profile likelihood function for ψ is given by $l_{p}\left(\psi ;y\right)=\underset{\lambda }{argmax}l\left(\psi ,\lambda ;y\right)$.

### 2.1. Bayesian Posterior Distribution of the Parameter of Interest and Effect of the Induced Priors

We will start by describing the Bayesian solution to the problem of inference for ψ. As is the case with all Bayesian inferences, inference for ψ is based on its posterior distribution π(ψ|y)=∫f(y|ψ,λ)π(ψ)π(λ|ψ)dλ. If the likelihood function f(y|ψ,λ) is analytical, the computation of this posterior distribution is relatively simple. Various numerical procedures, such as acceptance-rejection or importance sampling, may be used to generate random variates from the posterior distribution. For general hierarchical models, however, the marginal distribution f(y|ψ,λ)=∫f(y|x,ψ,λ)g(x)dx, where x denotes the latent variables (unobserved quantities in general), involves an integral that is of the same dimensions as x, which could be quite large. The beauty of the class of algorithms collectively known as the Markov chain Monte Carlo algorithms is that they allow one to avoid the explicit evaluation of this integral. Even more importantly for the problem of inference for the parameter of interest ψ, the MCMC-based Bayesian solution does not require explicit knowledge of the mapping θ↔(ψ,λ). This is an enormous advantage given that finding such an explicit mapping can be a difficult, and sometimes an impossible problem in many practical situations. Additionally, such a mapping need not be unique. The Bayesian solution using an MCMC algorithm is as follows:A prior distribution is specified on the canonical parameters θ, say π(θ).Using any of the myriads of MCMC algorithms, random variates from the posterior distribution π(θ|y) are generated. Let us denote these by $\theta_{1},\theta_{2},\ldots ,\theta_{S}$, where *S* is the MCMC sample size.To obtain random variates from the posterior distribution π(ψ|y), one simply transforms variates $\theta_{1},\theta_{2},\ldots ,\theta_{S}$ to $\psi_{1},\psi_{2},\ldots ,\psi_{S}$ using the mapping $\psi_{i}=f\left(\theta_{i}\right)$. Notice that, for this transformation, one does not require the explicit function for the incidental parameters λ. This is a major advantage because ψ=f(θ) is dictated by scientific interest, but finding the remaining components λ that complete the one-to-one transformation can be arduous.Given these random variates $\left\{\psi_{1},\psi_{2},\cdots ,\psi_{S}\right\}$, one can easily obtain the Bayes estimator of ψ, the credible interval associated with it, and other quantities of interest.

The Bayesian solution is simple, elegant, and practical, except that it depends on the specification of the prior distribution. Even when the prior distribution on θ is chosen to be non-informative, there is no guarantee that the induced prior distribution π(ψ) is non-informative. In fact, it is guaranteed to be informative [2,4]. Bayesian inference for induced parameters is thus likely to be based on informative priors. Furthermore, it also depends on π(λ|ψ). It may not even be possible to know what this distribution looks like because of the difficulty of finding the explicit function of θ that maps onto λ when the dimension of θ is large and because the mapping is not necessarily unique. Thus, Bayesian inference for the parameter of interest may have unknown biases. Such biases can have substantial consequences in practice when this is used for decision making [5].

### 2.2. Profile Likelihood

A solution that is invariant to parameterization is the profile likelihood for the parameter ψ. When the likelihood function is analytical (or, if it can easily be computed at any value of the parameter), computing the profile likelihood function is simply a constrained optimization problem. However, for hierarchical models, because of the high-dimensional integration over the latent state variables, computing the likelihood function itself is a significant computational problem. Computing the profile likelihood adds to this complexity further. On the other hand, as we saw above, obtaining the posterior distribution for any specified function of the parameters is quite easy once the MCMC samples from π(θ|y) are available. Our goal is to harness the simplicity of the Bayesian computation to obtain the profile likelihood.

#### 2.2.1. Laplace Approximation of the Posterior Distribution

The posterior distribution of ψ can be written as
$$\pi \left(\psi |y\right)=\frac{\int exp\{l(\psi ,\lambda ;y)\}\pi (\psi ,\lambda )d\lambda }{\int exp\{l(\psi ,\lambda ;y)\}\pi (\psi ,\lambda )d\lambda d\psi }.$$

Let us assume that the model allows the Laplace approximation [21] of the integrals. Then, following [22], we can write the posterior distribution approximately as
$$\pi \left(\psi |y\right)\propto exp\left\{l\left(\psi ,{\hat{\lambda }}_{\psi };y\right)\right\}|-\frac{\partial^{2}}{\partial \lambda^{2}}l\left(\psi ,\lambda ;y\right)-\frac{\partial^{2}}{\partial \lambda^{2}}log\pi \left(\lambda |\psi \right){|}_{\lambda ={\hat{\lambda }}_{\psi }}^{0.5}\pi \left(\psi \right)\pi \left({\hat{\lambda }}_{\psi }|\psi \right),$$
where $\left(\psi ,{\hat{\lambda }}_{\psi }\right)$ denotes the location of the constrained maximum of the function l(ψ,λ;y)+logπ(ψ,λ) with respect to λ for a fixed value of ψ. Note that this is *not* necessarily equal to the constrained MLE of λ for a fixed ψ because it is also affected by the choice of the prior distribution.

Following [17], let us use (y,y) as the new data, employing a technique called data doubling. The log-likelihood function based on the doubled data is 2×l(ψ,λ;y). Hence, we can write the Laplace approximation of the posterior distribution of ψ with the doubled data as follows:$$\pi \left(\psi |y,y\right)\propto exp\left\{2l\left(\psi ,{\hat{\lambda }}_{\psi (2)};y\right)\right\}|-2\frac{\partial^{2}}{\partial \lambda^{2}}l\left(\psi ,\lambda ;y\right)-\frac{\partial^{2}}{\partial \lambda^{2}}log\pi \left(\lambda |\psi \right){|}_{\lambda ={\hat{\lambda }}_{\psi (2)}}^{0.5}\pi \left(\psi \right)\pi \left({\hat{\lambda }}_{\psi (2)}|\psi \right),$$
where ${\hat{\lambda }}_{\psi (2)}$ is the location of the constrained maximum of 2×l(ψ,λ;y)+logπ(ψ,λ). Depending on the strength of the data and the prior, this could be similar to or quite different from ${\hat{\lambda }}_{\psi }$.

The ratio of these two posterior distributions can be written as a product of three terms, as follows:$$\frac{\pi (\psi |y,y)}{\pi (\psi |y)}=Term1\times Term2\times Term3,$$
where
(1)$$Term1=exp(\left\{2l\left(\psi ,{\hat{\lambda }}_{\psi (2)};y\right)-l\left(\psi ,{\hat{\lambda }}_{\psi };y\right)\right\}$$
(2)$$Term2=\frac{|-2\frac{\partial^{2}}{\partial \lambda^{2}}l\left(\psi ,\lambda ;y\right)-\frac{\partial^{2}}{\partial \lambda^{2}}{log\pi \left(\lambda |\psi \right)|}_{\lambda ={\hat{\lambda }}_{\psi (2)}}^{0.5}}{|-\frac{\partial^{2}}{\partial \lambda^{2}}l\left(\psi ,\lambda ;y\right)-\frac{\partial^{2}}{\partial \lambda^{2}}{log\pi \left(\lambda |\psi \right)|}_{\lambda ={\hat{\lambda }}_{\psi }}^{0.5}}$$
(3)$$Term3=\frac{\pi ({\hat{\lambda }}_{\psi (2)}|\psi )}{\pi ({\hat{\lambda }}_{\psi }|\psi )}$$

We show that with a judicious choice of the prior distribution, Terms 2 and 3 (Equations (2) and (3) above) can be made nearly constant, making the ratio of two posterior distributions approximate the profile likelihood for ψ.

#### 2.2.2. The Transformation θ⟷(ψ,λ)
Is Explicit

Situations when the explicit transformation θ⟷(ψ,λ) is available arise when the parameter of interest happens to be a component of the canonical parameter vector. For example, consider a regression model. In this case, one might be interested in the effect of a covariate of interest on the response in the presence of other covariates. This corresponds to obtaining the profile likelihood for the corresponding regression coefficient. Surprisingly, in such a situation, it is not necessary to estimate the MLE of the entire parameter vector. We can obtain the profile likelihood and hence the profile likelihood estimator directly using the data doubling technique. The fact that one does not need to compute the MLE makes this an attractive alternative to the approach in [17], at least in these situations.

Let us choose a prior distribution on λ such that the components of λ are independent and, individually, flat on the majority of their support. In choosing such flat priors for parameters with infinite support, one has to be careful to make sure that the resultant posterior distribution is a proper distribution. One way to assure this in practice is to choose a large but finite support set for each parameter that encompasses most of its probable values. In our experience, the standard practice of choosing prior distributions that are proper distributions with large variances assures proper posterior distribution and that they are also flat on most of the range of the respective parameters. We also choose these to be independent of ψ. With such a choice, it is clear that $\frac{d}{d\lambda_{i}}log\pi \left(\lambda_{i}\right)\approx 0$ and $\frac{d^{2}}{d\lambda_{i}^{2}}log\pi \left(\lambda_{i}\right)\approx 0$. Given this, and by choosing a flat prior on ψ, the following results are obvious:It is easy to see that ${\hat{\lambda }}_{\psi (2)}\approx {\hat{\lambda }}_{\psi }$. We are solving identical estimating equations under the original data and the doubled data. Hence, Term 3 (Equation (Equation 3) above) is exactly equal to 1.As a matrix, $\frac{\partial^{2}}{\partial \lambda^{2}}log\pi \left(\lambda |\psi \right)\approx 0$. Hence, Term 2 (Equation (Equation 2) above) is exactly equal to 1.Term 1 (Equation (Equation 1) above) is the profile likelihood.

We will illustrate the application of this simplified algorithm in some of the examples in Section 3, where the transformation θ⟷(ψ,λ) is explicit.

#### 2.2.3. The Transformation θ⟷(ψ,λ)
Is Not Explicit

Let us assume that the statistical model satisfies the regularity conditions needed for the maximum likelihood estimator $\left(\hat{\psi },\hat{\lambda }\right)$ to be asymptotically Gaussian with mean (ψ,λ) and precision matrix (the inverse of the variance matrix) i(ψ,λ) [23]. These conditions, along with some conditions on the prior distribution, imply that the posterior distribution π(ψ,λ|y) converges to Gaussian distribution with a mean equal to the MLE $\left(\hat{\psi },\hat{\lambda }\right)$, and with variance matrix $i^{-1}\left(\hat{\psi },\hat{\lambda }\right)$ [24] as that sample size increases. Let us use this asymptotic distribution as the prior distribution for computing the posterior distribution π(ψ|y). It is well-known that, under regularity conditions, the following results hold [23]:For a large enough sample size, the log-likelihood function is well approximated by the quadratic $l\left(\psi ,\lambda ;y\right)\approx constant-0.5\left(\psi -\hat{\psi },\lambda -\hat{\lambda }\right)$. $i\left(\hat{\psi },\hat{\lambda }\right){\left(\psi -\hat{\psi },\lambda -\hat{\lambda }\right)}^{T}$By choice, the log-prior distribution is $log\pi \left(\psi ,\lambda \right)=constant-0.5\left(\psi -\hat{\psi },\lambda -\hat{\lambda }\right)i\left(\hat{\psi },\hat{\lambda }\right)$${\left(\psi -\hat{\psi },\lambda -\hat{\lambda }\right)}^{T}$.It follows that $l\left(\psi ,\lambda ;y\right)+log\pi \left(\psi ,\lambda \right)\approx constant-\left(\psi -\hat{\psi },\lambda -\hat{\lambda }\right)i\left(\hat{\psi },\hat{\lambda }\right){\left(\psi -\hat{\psi },\lambda -\hat{\lambda }\right)}^{T}$.

The location of the maximum of λ for a given value of ψ in the quadratic function is given by ${\hat{\lambda }}_{\psi }=\hat{\lambda }+{\left(i^{\hat{\lambda }\hat{\lambda }}\right)}^{-1}i^{\hat{\lambda }\hat{\psi }}\left(\psi -\hat{\psi }\right)$. This is approximately the mode of the posterior distribution of ψ if the quadratic approximation of the log-likelihood is reasonable.

Following similar calculations with doubled data, the following is true:$$l\left(\psi ,\lambda ;y,y\right)+log\pi \left(\psi ,\lambda \right)\approx 1.5\left(\psi -\hat{\psi },\lambda -\hat{\lambda }\right)i\left(\hat{\psi },\hat{\lambda }\right){\left(\psi -\hat{\psi },\lambda -\hat{\lambda }\right)}^{T}.$$
It follows that the posterior mode ${\hat{\lambda }}_{\psi (2)}\approx {\hat{\lambda }}_{\psi }$.Hence, Term 3 (Equation (Equation 3) above) is approximately 1.It also follows that Term 1 (Equation (Equation 1) above) is approximately $l_{p}\left(\psi ;y\right)$.Given the quadratic approximation, it follows that the negative second derivative (Hessian matrix) is approximately
$$-\frac{\partial^{2}}{\partial \lambda^{2}}l\left(\psi ,\lambda ;y\right)-\frac{\partial^{2}}{\partial \lambda^{2}}log\pi \left(\lambda |\psi \right)\approx 2\times i_{\lambda \lambda }\left(\psi ,\lambda \right)$$
and
$$-2\frac{\partial^{2}}{\partial \lambda^{2}}l\left(\psi ,\lambda ;y\right)-\frac{\partial^{2}}{\partial \lambda^{2}}log\pi \left(\lambda |\psi \right)\approx 3\times i_{\lambda \lambda }\left(\psi ,\lambda \right).$$Hence, Term 2 (Equation (Equation 2) above) is approximately $\sqrt{3/2}$.

Essentially, any prior that matches, at least approximately, the first and second derivatives (with respect to λ, for a fixed ψ) of the log-posterior distribution with those of the log-likelihood function will work. For example, one can use a flat distribution for π(λ|ψ) provided an explicit transformation (ψ,λ) is available.

We can obtain the MLE and its asymptotic variance for the parameters of a hierarchical model using any preferred method. Hierarchical models are too complex to analytically prove the estimability of the parameters and the rate of convergence of the estimators. The method of data cloning has a built-in diagnostic step for the non-estimability of the parameters and can be used to check if the convergence is at the appropriate rate of the square root of the sample size [25]. Moreover, existing programs for data cloning can be used, unmodified, to generate random variates from the posterior distribution with doubled data, as we will see in our examples.

### 2.3. Description of the Algorithms

We now describe the Algorithms 1 and 2 explicitly.    
**Algorithm 1:** Algorithm when θ⟷(ψ,λ) is explicit
Step 1: Specify an (essentially) flat prior for each of the components of (ψ,λ) such that they are independent of each other.Step 2: Generate random variates from π(ψ,λ|y) and pick the first component. Denote this by $\left\{\psi_{1}^{\left(1\right)},\psi_{2}^{\left(1\right)},\cdots ,\psi_{S}^{\left(1\right)}\right\}$.Step 3: Generate random variates from π(ψ,λ|y,y) and pick the first component. Denote this by $\left\{\psi_{1}^{\left(2\right)},\psi_{2}^{\left(2\right)},\cdots ,\psi_{S}^{\left(2\right)}\right\}$.Step 4: Create a new data matrix
$$M^{T}=\left[\begin{array}{cccccccc} 1 & 1 & \cdots & 1 & 0 & 0 & \cdots & 0\\ \psi_{1}^{\left(2\right)} & \psi_{2}^{\left(2\right)} & \cdots & \psi_{S}^{\left(2\right)} & \psi_{1}^{\left(1\right)} & \psi_{2}^{\left(1\right)} & \cdots & \psi_{S}^{\left(1\right)} \end{array}\right].$$Step 5: Conduct a logistic regression analysis on this data matrix with responses as the first column and covariate values as the second column.Step 6: The log-odds function, ignoring the intercept, is the estimate of the log-profile likelihood.


    Notice that steps 4–6 correspond to estimating the logistic-regression-based classification rule to discriminate between two populations, where the classification rule is based on the ratio of two densities [26]. A general additive model (GAM) may also be used for the link function if more flexibility is required. The GAMs tend to be fragile in the tails, so we suggest caution in using these flexible models. One may also use any machine learning approaches to estimate the classification rule between two populations. Similarly, one can simply use any non-parametric density estimator to obtain $\hat{\pi }\left(\psi |y,y\right)$ and $\hat{\pi }\left(\psi |y\right)$ and take the difference in the log-densities. In our experience, a simple logistic regression with a quartic log-odds function or GAM with a concavity constraint works quite well.
**Algorithm 2:** Algorithm when θ⟷(ψ,λ) is *not* explicitStep 1: Compute the MLE $\hat{\theta }$ and its asymptotic variance $i^{-1}\left(\hat{\theta }\right)$ for the model with canonical parameters.Step 2: Use the asymptotic normal distribution, $\pi \left(\theta \right)=N(\hat{\theta },i^{-1}\left(\hat{\theta }\right))$, as the prior and use MCMC to generate random variates from π(θ|y) and π(θ|y,y).Step 3: Given the random variates in Step 2, use the transformation f(θ)=ψ to obtain $\left\{\psi_{1}^{\left(1\right)},\psi_{2}^{\left(1\right)},\cdots ,\psi_{S}^{\left(1\right)}\right\}$ and $\left\{\psi_{1}^{\left(2\right)},\psi_{2}^{\left(2\right)},\cdots ,\psi_{S}^{\left(2\right)}\right\}$. Although we are using the same number of random variates *S* for both posterior distributions, this is not needed.Step 4: Create a new data matrix
$$M^{T}=\left[\begin{array}{cccccccc} 1 & 1 & \cdots & 1 & 0 & 0 & \cdots & 0\\ \psi_{1}^{\left(2\right)} & \psi_{2}^{\left(2\right)} & \cdots & \psi_{S}^{\left(2\right)} & \psi_{1}^{\left(1\right)} & \psi_{2}^{\left(1\right)} & \cdots & \psi_{S}^{\left(1\right)} \end{array}\right].$$Step 5: Conduct a logistic regression analysis on this data matrix with responses as the first column and covariate values as the second column.Step 6: The log-odds function, ignoring the intercept, is the estimate of the log-profile likelihood.


*Notes*: (1) We note that once we have the output from Step 2, we can compute profile likelihood for any function of interest easily. No additional runs of the MCMC algorithm are required. On the other hand, even when analytical likelihood function is available, each different function f(θ) requires a separate run of the constrained optimization routine. This suggests that the above algorithm might be a good competitor even when the likelihood function is available analytically.

(2) When applying the MCMC algorithm for generating these samples, we suggest that a fairly large number of chains (with possibly a small number of iterates from each chain) be used. In our experience, this usually leads to a better coverage of the tail areas of the posterior distributions than with one or a small number of chains.

(3) One can also use an approximation to the posterior densities π(ψ|y,y) and π(ψ|y) as described in INLA [11] and take the ratio directly without using the logistic regression approach described above. In situations where INLA is appropriate, this is likely to be computationally faster than running the MCMC algorithm.

## 3. Examples

We start with a few situations where the log-profile likelihood can be computed analytically. We compare the data-doubling-based log-profile likelihood with the analytical one to show that the method works in practice. We then consider hierarchical models where the analytical log-profile likelihood is difficult to compute. In these cases, when appropriate, we compare the performance of the two algorithms described above.

We used the R package dclone [27] to conduct these simulations. We used JAGS [9] or Stan [10], as appropriate, to conduct the MCMC component of the method. We checked the MCMC convergence and mixing using the Gelman–Rubin ’Rhat’ statistics (see [9] for details) and trace plots. We used the R package mgcv [28] to conduct the logistic regression to estimate the log-profile likelihood function. We used three different smoothers: a quartic function, spline smoother, and spline smoother with a concavity constraint. These three smoothers tend to give very similar answers. To avoid clutter in the figures, we only plotted the profile likelihood based on the concavity constraint. The complete code for each example is available from the author upon request and is also available on the https://github.com/sublele/DDProfile-likelihood-code (accessed on 18 June 2023). In the figures, a solid black line indicates the analytical log-profile likelihood, a dashed line is the log-profile likelihood based on Algorithm 2, and a double-dashed line is the estimator based on Algorithm 1.

**Example 1.** 
*Beta distribution.*


We start with a simple example of computing the log-profile likelihood functions for features of a beta distribution with canonical parameters (α,β) and PDF $f\left(y;\alpha ,\beta \right)=\frac{y^{\alpha -1}{\left(1-y\right)}^{\beta -1}}{B(\alpha ,\beta )},0<y<1$ and α>0,β>0. Let the parameters of interest be mean, variance, and skewness: $\psi_{1}=\frac{\alpha }{\alpha +\beta }$, $\psi_{2}=\frac{\alpha \beta }{{\left(\alpha +\beta \right)}^{2}\left(\alpha +\beta +1\right)}$, and $\psi_{3}=\frac{2\left(\beta -\alpha \right)\sqrt{\left(\alpha +\beta +1\right)}}{\left(\alpha +\beta +2\right)\sqrt{\alpha \beta }}$, respectively. Notice that, except for the mean, the appropriate incidental parameter λ for each of these transformations is not a simple function of the canonical parameters (α,β). Furthermore, note that the transformation θ⟷(ψ,λ) is not necessarily unique. For example, one can consider $\left(\alpha ,\beta \right)⟷\left(\psi_{1},\alpha \right)$, $\left(\alpha ,\beta \right)⟷\left(\psi_{1},\beta \right)$, or $\left(\alpha ,\beta \right)⟷\left(\psi_{1},\alpha +\beta \right)$.

We generated *N* = 10,50 observations from a beta distribution with α=3,β=2. In the first row of Figure 1, we plot the estimate of the log-profile likelihood for $\psi_{1}$ using both algorithms. Recall that for Algorithm 1, we do not need to compute the MLEs of the canonical parameters, whereas for Algorithm 2, we use the asymptotic distribution of the MLE as the prior distribution. We compare the performance of these two algorithms with the analytical log-profile likelihood.

To check how DD works for more complex situations, we use the same data as above but compute the log-profile likelihood for $\psi_{2}$ and $\psi_{3}$. In these cases, the transformation θ⟷(ψ,λ) is not easy to compute. Hence, we only use Algorithm 2 and compare it with the analytical log-profile likelihood. These results are presented in the last two rows of Figure 1. It is quite clear that the data doubling algorithms work quite well for estimating the true log-profile likelihood for small and large sample sizes.

**Example 2.** 
*Disease dynamics model.*


We now consider a somewhat more complex situation of modeling disease transmission in a closed population. One of the simplest models for disease transmission is the compartmental SIR (susceptible-infected-recovered) model. We consider data on an outbreak of influenza A (H1N1) in 1978 at a British boarding school. The data consist of the daily number of students in bed, spanning a time interval of 14 days. There were 763 male students who were mostly full-boarders, and 512 of them became ill. The outbreak lasted from the 22nd of January to the 4th of February. It was reported that one infected boy started the epidemic, which spread rapidly in the relatively closed community of the boarding school. The data are freely available in the R package ‘outbreaks’. (for details, see https://rpubs.com/choisy/sir (accessed on 21 August 2023)). An important parameter of interest for these models is the so called basic reproduction rate $R_{0}$ [13]. This model can be parameterized in terms of infection rate β and recovery rate γ or equivalently in terms of $R_{0}=\frac{\beta }{\gamma }$ and γ. Hence, we can use both algorithms described in this paper. In addition, we can also compute the log-profile likelihood for $R_{0}$ analytically. In Figure 2, we present the log-profile likelihood estimates using the two algorithms along with the analytical log-profile likelihood. It is clear that the methodology described above indeed works. While computing the analytical log-profile likelihood, we noticed that the results were sensitive to the choice of the starting values. An important computational advantage of DD over numerical optimization techniques is that it does not require such starting values.

**Example 3.** 
*Multivariate normal distribution.*


In the analysis of multivariate data, the maximum eigenvalue of the covariance matrix is of interest. This is clearly a very non-linear function of the underlying parameters. We re-analyze the data, which were also used in [17], on five test scores for N = 88 students [29]. The data are assumed to have come from five-dimensional multivariate normal distribution with 20 canonical parameters (5 for the means and 15 for the variances and covariances). The parameter of interest is the largest eigenvalue of the covariance matrix. To facilitate MCMC convergence, we considered only three test scores (D = 3) instead of all five and scaled the data. However, in the analysis, we did not assume that the mean vector was 0 or that the variances were 1, thus keeping the same number of canonical parameters. The resulting profile likelihoods, one based on DD and one analytical, are presented in Figure 3b. Before conducting the real data analysis, we also conducted an analysis on simulated data from a multivariate normal distribution with dimensions D = 2 and sample size N = 50 to check if DD works when the model is guaranteed to be valid. Figure 3a presents the log-profile likelihood under simulated data, and Figure 3b presents the log-profile likelihood for the test score data. In this case, we could only use Algorithm 2, and hence the comparison was between Algorithm 2 and the analytical log-profile likelihood. Clearly, DD worked quite well even in this complex situation. As in the previous examples, we noticed non-convergence issues for the numerical optimization routine in calculating the analytical log-profile likelihood, especially for values that were farther away from the center.

**Example 4.** 
*Non-linear time series with measurement error.*


We re-analyze the classic population growth timeseries data on *Paramecium aurelia* [30] also used by [19]. Following these authors, we use the Beverton–Holt population dynamics model with Poisson measurement error. This is an example of a hierarchical model. We first compute the log-profile likelihood for the growth parameter λ. The Beverton–Holt model with Poisson measurement error has three parameters, $(\lambda ,\beta ,\sigma^{2}).$ The parameter of interest, λ, is the growth parameter. This is a component of the canonical parameter vector. Hence, we apply both algorithms to compute the log-profile likelihood. Figure 4a shows that the two algorithms lead to very similar log-profile likelihoods. Now, suppose we are also interested in another important quantity, $K=\frac{\lambda -1}{\beta }$, the carrying capacity of the system. Using the computational output already available, we can compute the log-profile likelihood for the carrying capacity by applying Algorithm 2. This is presented in Figure 4b. Alternatively, we can rewrite the model in terms of $\left(K,\beta ,\sigma^{2}\right)$, making *K* a component of the canonical parameter vector. Hence, we compute the log-profile likelihood for *K* using Algorithm 1. These two approaches match each other quite well. The advantage of Algorithm 1 over Algorithm 2 is that we do not have to estimate the MLE of the canonical parameters. This is a also a major advantage over the algorithms presented in [19,20] that require the initial estimation of the MLE.

**Example 5.** 
*Binary regression with random intercepts.*


Generalized linear mixed models, a widely used class of hierarchical models, are useful in many practical situations [31]. As an example, [32] considers pooling information across 22 clinical trials to assess the effect of beta blockers to prevent mortality after miocardial infarction using a random effects binary regression model. Each clinical trial has $n_{i}^{c}$ and $n_{i}^{t}$ individuals in the control and treatment groups, respectively, with i=1,2,…,22. The model components can be described as: $r_{i}^{c}∼binomial\left(n_{i}^{c},p_{i}^{c}\right)$ and $r_{i}^{t}∼binomial\left(n_{i}^{t},p_{i}^{t}\right)$, where $logit\left(p_{i}^{c}\right)=\mu_{i}$, $logit\left(p_{i}^{t}\right)=\mu_{i}+\delta_{i}$, $\delta_{i}∼N\left(\delta ,\tau_{1}^{-1}\right)$, and $\mu_{i}∼N\left(\mu ,\tau_{2}^{-1}\right)$.

The parameter of interest is the common treatment effect δ. Figure 5 presents the log-profile likelihood for the common log-odds ratio using both algorithms. They match each other quite well.

## 4. Discussion

We showed that once the MLE and its asymptotic variance is available, the profile likelihood for any parameter of interest can be easily computed using the method of data doubling. For our illustrations, we used the method of data cloning to obtain the MLE and its asymptotic variance. One important reason for this was convenience. An R package ’dclone’ [27] conveniently allowed us to obtain the MLE, its asymptotic variance of the canonical parameters, and the posterior distributions under data doubling. Although computationally demanding, the method of data cloning has a few advantages that make the computational effort worthwhile. By choosing the number of clones judiciously, it allows one to utilize the built-in smoothing of the Bayesian approach. This feature makes it suitable for ill-behaved, possibly multimodal likelihoods, at the same time reducing the effect of the prior specification. Data cloning is a global optimization algorithm that converges to the true mode of the likelihood function provided the full parameter space is explored effectively [8,33]. Hierarchical models, being inherently complex, face the possibility of the non-estimability of the canonical parameters. Data cloning has a built-in mechanism that allows one to diagnose the non-estimability of the canonical parameters during the course of implementation [25]. More interestingly, one can also use DC to diagnose the estimability of a specified function of the canonical parameters [25,34].

We briefly show that data doubling methodology may also be used to obtain the MLE, but not the asymptotic variance, of any general hierarchical model. Let $\theta =(\theta_{1},\theta_{2},\ldots ,\theta_{d})$. We can apply Algorithm 1 repeatedly to obtain the MLE profile of ${\hat{\theta }}_{i}$ for i=1,2,…,d by choosing flat and independent priors for each of the components. Noting that the MLE profile is also the MLE of the corresponding parameter, the vector $\hat{\theta }=\left({\hat{\theta }}_{1},{\hat{\theta }}_{2},\ldots ,{\hat{\theta }}_{d}\right)$ is the MLE of θ. Thus, one can obtain the MLE of θ with just two clones. This method, implicitly, uses the divide-and-conquer approach described in [35]. The usual method of data cloning uses a large number of clones. This can run into problems with the convergence of the MCMC algorithms, because the likelihood function becomes too concentrated. The use of only two clones avoids these MCMC convergence issues to a substantial extent. However, data doubling assumes that posterior distribution can be well-approximated by the Laplace approximation, whereas DC works in more general situations. A thorough discussion of these pros and cons along with the application of the methodology to non-likelihood-type objective functions will be provided in a separate paper.

A major difficulty in computing the profile likelihood for hierarchical models is that the likelihood function is, either analytically or computationally, difficult to obtain. However, one can use the fact that the log-likelihood function can be approximated using the Taylor series expansion when the MLE and Fisher information matrix are available. One can, potentially, use this quadratic approximation to obtain an approximate log-profile likelihood for any function of a parameter using constrained optimization. Although this might be a good approximation in many cases, it is obvious that such a profile likelihood function cannot be parameterization-invariant. Let us consider the beta distribution example discussed earlier. One can parameterize the model using either natural parameterization in terms of α and β or in terms of $\psi_{1}$ and α. Suppose that we are interested in the profile likelihood for $\psi_{1}$. Under the second parameterization and the quadratic approximation to the log-likelihood, the log-profile likelihood for $\psi_{1}$ is a quadratic function, whereas the log-profile likelihood under natural parameterization, the log-profile likelihood for $\psi_{1}$, is not a quadratic function (although it may be close to being quadratic). One interesting idea that we plan to explore in a future publication is to combine the DD-based log-profile likelihood with that obtained using the quadratic approximation to the log-likelihood, as described above. This may combine the smoothness inherent in the quadratic approximation with the parameterization invariance of the DD-based log-profile likelihood; the resultant log-profile likelihood might be better-behaved than either the DD- or quadratic-approximation-based log-profile likelihood.

The results in this paper are relevant when Laplace approximation is applicable. However, there are a class of problems under the umbrella of non-standard asymptotics where Laplace approximation may not be applicable. For example, see [36,37] or [38]. Extending the data doubling methodology to these problems is an important component of our future research projects.

## Figures and Tables

**Figure 1 entropy-25-01262-f001:**
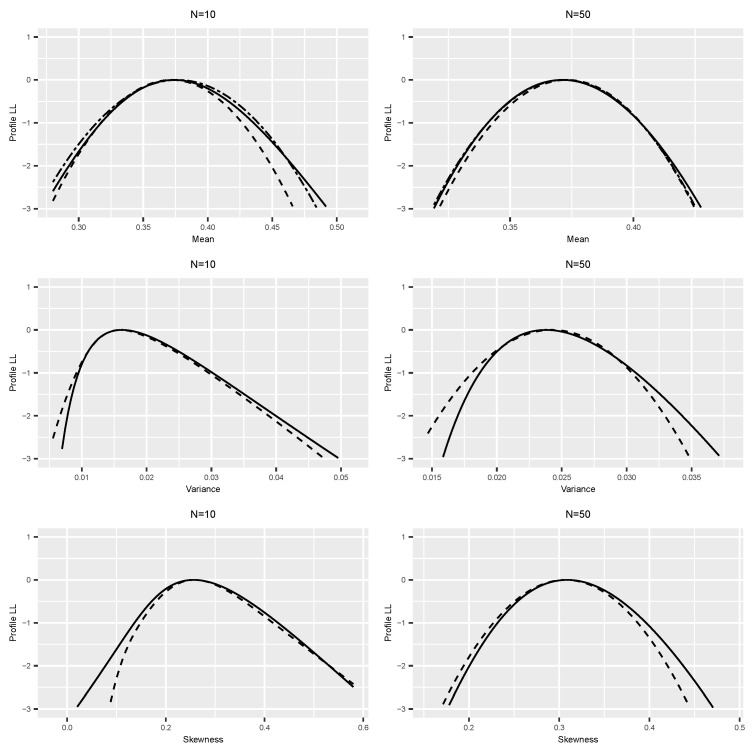
Log-profile likelihoods under beta (3,5) distribution for mean, variance, and skewness with sample sizes 10 and 50. Solid black line = analytical log-profile likelihood, dashed line = Algorithm 2, double-dashed line = Algorithm 1.

**Figure 2 entropy-25-01262-f002:**
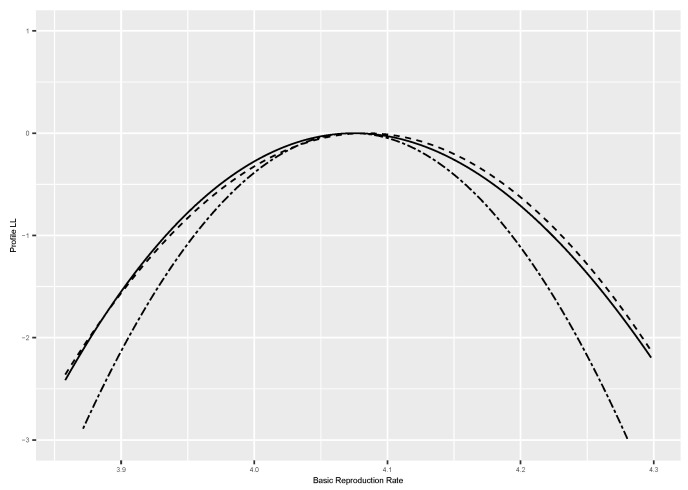
Log-profile likelihood of basic reproduction rate (R0) for the SIR model. Solid black line = analytical log-profile likelihood, dashed line = Algorithm 2, double-dashed line = Algorithm 1.

**Figure 3 entropy-25-01262-f003:**
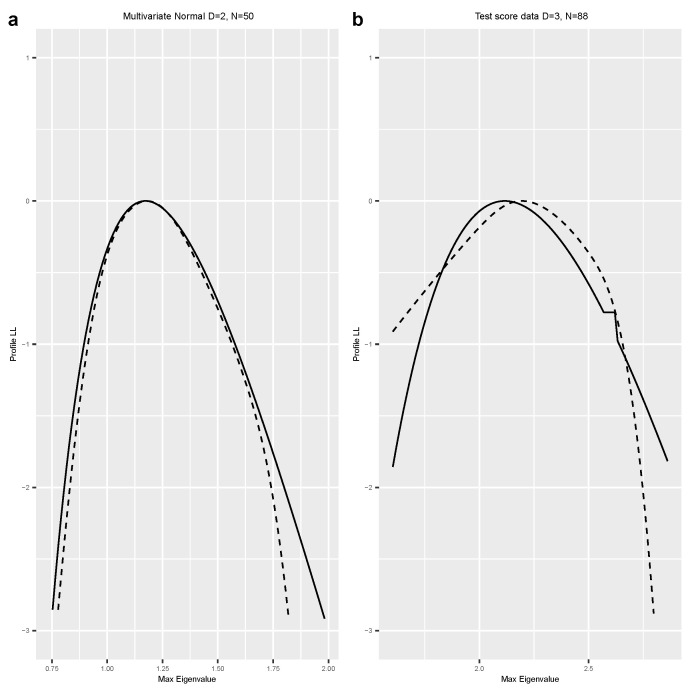
Log-profile likelihood of maximum eigenvalue for multivariate normal distribution (simulated data in (**a**) and test score data in (**b**)). Solid black line = analytical log-profile likelihood, dashed line = Algorithm 2.

**Figure 4 entropy-25-01262-f004:**
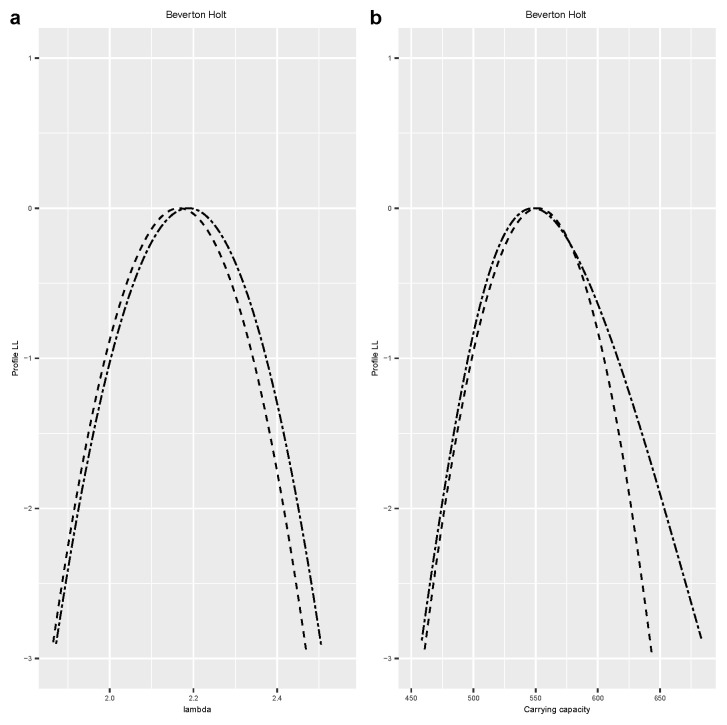
Log-profile likelihood of growth and carrying capacity for the Beverton–Holt model. (**a**) Lambda: the growth parameter; (**b**) log-profile likelihood for the carrying capacity. Dashed line = Algorithm 2, double-dashed line = Algorithm 1.

**Figure 5 entropy-25-01262-f005:**
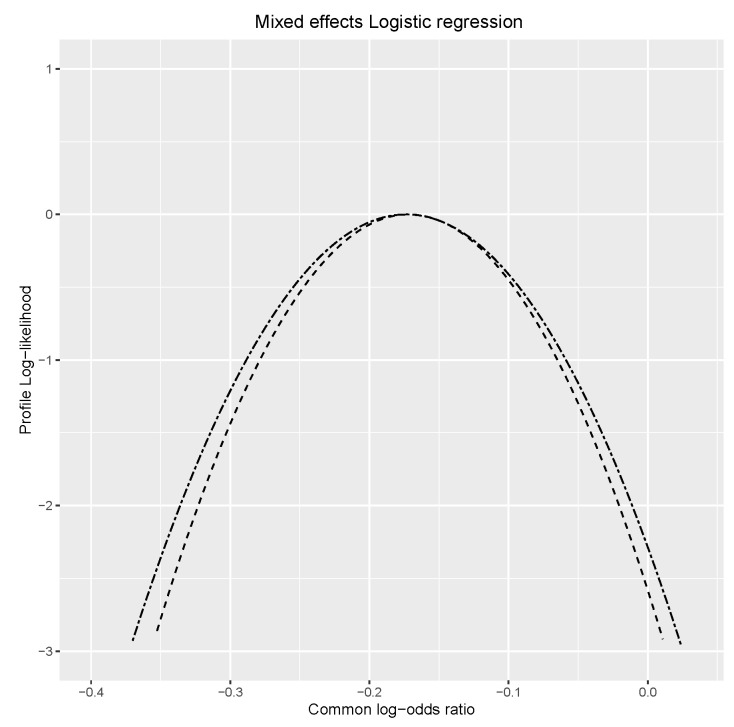
Log-profile likelihood of common log-odds ratio for logistic regression with random effects. Dashed line = Algorithm 2, double-dashed line = Algorithm 1.

## Data Availability

https://github.com/sublele/DDProfile-likelihood-code (accessed on 18 June 2023).

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
