# Peer review of "Profile Likelihood for Hierarchical Models Using Data Doubling"

_entropy, 2023, doi:10.3390/e25091262_

Round 1
Reviewer 1 Report
The manuscript entitled “Profile likelihood for hierarchical models using data doubling” presented an interesting proposal to compute profile likelihood for any specified function of the parameters of hierarchical models using data doubling and I fell that it has merit and can be published in ENTROPY. However the manuscript still requires some minor revisions.
Here are some aspects that need review:
- line 16: The introduction is not numbered. Consequently, the numbering of the other sections should be updated.
- line 135: The author could number the posteriori distribution in that of line 143 and then reference that equation in line 145: "integral in the numerator (X)..."
- lines 152 and 155: exclude "*" from math expressions
- line 158-159: Adopt "\times" instead of "*"
- lines 178-180,203,204 and 208: Since the equations for Terms 1, 2 and 3 have been numbered (line 159), include their references when quoting them in the text.
- Sections 1.3.1 and 1.3.2: To number the algorithms (Algorithm 1 and Algorithm 2), since they have been quoted in the text.
- Algorithm 1: The author could include some discussion about "Step 1" of Algorithm 1. For example: if the parameter is unlimited, then is the flat prior improper? In this case it must be guaranteed that the posteriori is proper. Some discussion about the fact that Step 1 can actually consider other prioris besides the flat prior.
- line 277: JAGS reference is missing
- line 279: What is “Rhat”?
- line 284: What is “XXX”?
- title of all figures: Isn't the correct "profile log-likelihood"? Revise the entire text of the manuscript (e.g., in line 298: "In the first line of Figure 1, we plot the profile LOG-likelihood estimate...).
- line 298: typo: “FIgure 1”
- line 340: describe what is D=2 and N=50 (size and sample size?)
- line 358: typo: “ALgorithm”
- Figure 3a Why is the profile log-likelihood constant after a certain value?
- Figure 3b was not commented and discussed in the text. How does the result differ from figure 3a? What is the motivation for considering N=88?
- Figure 5: Figure 5 is displayed in the text before example 5
- line 400: The author should include some discussion of the consequences of DD, especially with regard to the precision of the estimates. Doesn't DD artificially increase the sample size and therefore (artificially) increase the precision of the estimates?
Author Response
Thank you for your kind and useful review. I am attaching a file that addresses your comments in detail.

Reviewer 2 Report
Title: Profile likelihood for hierarchical models using data doubling
By: Subhash Lele
Submitted to: Entropy, Ms I.d. Entropy-2484862
Report 07/05/2023
The author describe a simple computational method to compute profile likelihood for any
specified function of the parameters of a general hierarchical model using data doubling,
and provides a mathematical proof for the validity of the method under regularity
conditions.
Major Comments:
* The profile likelihood is mainly designed to estimate a finite dimensional parameter
of interest in the presence of an infinite dimensional nuisance parameter.
Here dim(\theta)=d, and \thet(\psi,\lambda), so the nuisnace parameter \lambda is an
finite dimensional parameter, and the profile likelihood
l_p(\psi|y) = \max_\lambda l(\psi,\lambda|y)
is trivial.
* The author used "data doubling", or use (y,y) as the new data based on the
original data y. In terms of the original log-likelihood l(\psi,\lambda|y), the new
log-likelihood is now
l(\psi,\lambda|y,y)= 2*l(\psi,\lambda|y).
Why this helps in the estimation? Intuitively for any $C \= 0, using
C*l(\psi,\lambda|y) and l(\psi,\lambda|y) are the same in estimation, please explain.
How about use data "k-pling" (y,...y), and the log-likehood
l(\psi,\lambda|y,...,y)= k*l(\psi,\lambda|y)?
* The asymptotic approximation (Laplace approximation) is used in the computation of
estimation, via the Bayesian method, which is often used in case of small sample
size to incorporate the prior information. Here if we assume large sample size, the
Bayesian and Frequentist approaches are asymptotically equivalent (The Bernsterin-von Misese theorem), and the frequentist method is much simpler to use.
Moderate improvement needed
Author Response
Thank you for your comments on my manuscript. I am attaching a file that provides responses to your comments in detail. I hope you find them satisfactory.

Reviewer 3 Report
Reviewer’s Report on Ms Entropy_2484862:
“Profile likelihood for hierarchical
models using data doubling”
by Subhash Lele
The paper proposes a method to compute the profile likelihood of a hierarchical model using data doubling. The paper theoretically motivates the method under some regularity conditions ensuring that the asymptotic distribution of the maximum likelihood estimator is non-singular multivariate Gaussian.
As remarked by the author, an appropriate statistical model often involves many parameters, while interest is limited to their scalar functions. In frequentist inference, the profile likelihood provides the default approach to the problem, but it can be computationally challenging for hierarchical models.
The paper proposes a method for computing the profile likelihood which appears to be very promising. Its merits are illustrated with some models. Unfortunately, some of these models are not the hierarchical ones whose hierarchical likelihood poses more problems.
As a first example, consider nonlinear time series, which are also mentioned in the paper. However, the example is of limited interest, and it is interesting only for illustrative purposes. The nonlinear time series posing more problems, and which are most investigated, are the financial ones.
ARCH models (Engle, 1982), GARCH models (Bollerslev, 1986) and their generalizations have a hierarchical structure. Their (profile) likelihoods do not have a closed form. The problem gets more complicated when the error is not normal but is skew normal or a mixture of skew-normals (De Luca and Loperfido, 2004).
I understand that the problems posed by GARCH-type models fall outside the scope of the present paper, but the author should at least mention them as directions of future research. Otherwise, the method proposed in the paper might appear of very limited use and therefore of little interest.
Bollerslev, T. (1986). Generalized autoregressive conditional heteroskedasticity. Journal of Econometrics 31, 307–327.
De Luca, G. and Loperfido, N. (2004). A Skew-in-Mean GARCH Model for Financial Returns. In “Skew-Elliptical Distributions and Their Applications: A Journey Beyond Normality”, CRC/Chapman & Hall, 205-222.
Engle, R. F. (1982). Autoregressive conditional heteroskedasticity with estimates of the variance of UK inflation. Econometrica 50, 987–1008.
Author Response
Thank you for your comments on my manuscript. I am attaching a file that details my responses. I hope you find them satisfactory.

Round 2
Reviewer 1 Report
I believe that the author have made the corrections and answered the questions satisfactorily. Thus, I believe that the manuscript can be published in ENTROPY.
In addition, regarding the questioning: "The author should include some discussion of the consequences of DD, especially with regard to the precision of the estimates. Doesn't DD artificially increase the sample size and therefore (artificially) increase the precision of the estimates?", I believe it is interesting to include the explanation in the paper.
Reviewer 2 Report
The proposed method is trivial without enough new contribution.
Reviewer 3 Report
You satisfactorily addressed all my remarks.